# Linking traits based on their shared molecular mechanisms

**Yael Oren, Aharon Nachshon, Amit Frishberg, Roni Wilentzik, Irit Gat-Viks\***

Department of Cell Research and Immunology, George S. Wise Faculty of Life Sciences, Tel Aviv University, Tel Aviv, Israel

**Abstract** There is growing recognition that co-morbidity and co-occurrence of disease traits are often determined by shared genetic and molecular mechanisms. In most cases, however, the specific mechanisms that lead to such trait–trait relationships are yet unknown. Here we present an analysis of a broad spectrum of behavioral and physiological traits together with gene-expression measurements across genetically diverse mouse strains. We develop an unbiased methodology that constructs potentially overlapping groups of traits and resolves their underlying combination of genetic loci and molecular mechanisms. For example, our method predicts that genetic variation in the *Klf7* gene may influence gene transcripts in bone marrow-derived myeloid cells, which in turn affect 17 behavioral traits following morphine injection; this predicted effect of *Klf7* is consistent with an in vitro perturbation of *Klf7* in bone marrow cells. Our analysis demonstrates the utility of studying hidden causative mechanisms that lead to relationships between complex traits.

**\*For correspondence:** iritgv@post.tau.ac.il

**Competing interests:** The authors declare that no competing interests exist.

## Introduction

Epidemiological and clinical research has identified a profusion of correlated physiological traits, as well as a remarkably high incidence of co-occurrence and comorbidity among disorders. Various studies have shown that such connections among diseases are typically attributable to a common underlying genetic or molecular mechanism (*Rzhetsky et al., 2007*; *Oti et al., 2008*; *Barabási et al., 2011*; *Cotsapas et al., 2011*; *Lee et al., 2012*; *Cross-Disorder Group of the Psychiatric Genomics Consortium, 2013*). Disclosure of unexpected relationships among disease phenotypes and understanding of their common mechanisms opens the way to improved disease classification and treatment. In particular, it may allow a drug approved for one disease to be used for the treatment of another disease, and provide us with the means to anticipate undesired off-target effects (e.g., *Ashburn and Thor, 2004*; *Dudley et al., 2011*).

Advanced computational methods have made it possible to study the mechanisms underlying trait connections in an unbiased manner. One approach is to derive trait connectivity based on trait–trait comorbidity, co-occurrence, and correlations (*Hidalgo et al., 2009*; *Shi et al., 2010*; *Blednov et al., 2012*). As an example, *Figure 1—figure supplement 1A* illustrates a sample network and *Figure 1—figure supplement 1B* depicts a group of correlated traits in this network. Relying entirely on trait information, however, makes it difficult to identify the shared mechanisms and to distinguish shared molecular mechanisms from shared environmental influences. Alternatively, a common way to improve predictions is by integrating relationships between genes and traits, using gene–trait correlations, associations, or causal mutations (*Rzhetsky et al., 2007*; *Cotsapas et al., 2011*; *Baker et al., 2012*; *Hwang et al., 2012*; *Gat-Viks et al., 2013*). Such pairwise gene–trait connections were used to construct two-layer clusters ('biclusters') consisting of groups of traits linked to the same group of genes. For example, *Figure 1—figure supplement 1C* depicts a bicluster for the sample network in *Figure 1—figure supplement 1A*. Notably, although such 'gene-based' approaches provide a list of putative non-environmental mechanisms, their utilization has two major

**eLife digest** Many individuals who have diabetes also have other diseases that affect the heart and blood vessels. It is not uncommon for human diseases to occur together like this; and understanding the relationships between diseases and other traits can make it easier to diagnose conditions. Furthermore, it can also help researchers develop treatments that are more precisely targeted to each condition and cause fewer side effects.

Two conditions or traits tend to occur together if they are caused by mutations in the same gene or genes; or if they involve processes within cells that share the same proteins and other molecules. However, in most cases the genes and molecular mechanisms involved are not yet known so it is more difficult to work out how the traits are connected.

Computing techniques make it possible to assess the relationships between hundreds or thousands of traits at the same time. These high volume analyses can also allow scientists to identify less obvious relationships that might be missed in more traditional types of study.

Here, Oren et al. created a new computer algorithm to identify related traits, their shared genetic basis, and the molecular mechanisms behind them. The algorithm is called GEMOT and uses a three-step approach to sift through a large amount of data. Oren et al. tested GEMOT using a database of 1738 documented traits—including diseases and behaviors—in laboratory mice.

Oren et al. identified many clusters of traits in the mice and the underlying genetic and molecular mechanisms that link them. For example, they found that a mutation in a gene called *Klf7* affected the expression of other genes that are involved in making new cells in the bone marrow. In turn, these changes influenced 17 different behaviors in the mice after they were injected with the painkiller morphine.

In humans, the same genes that underlie behaviors related to morphine treatment have been linked to the survival rate of patients with a form of brain cancer. This suggests that—alongside providing pain-relief—morphine may influence how the tumor grows. The algorithm developed by Oren et al. can now be used to further explore the impact of the environment on the relationships between traits.

drawbacks. First, these approaches assign a single mechanistic layer whereas in fact what is affected by genetic variation is a number of molecular components (such as transcripts), which affect the physiological traits secondarily; thus, the model should include a series of mechanistic layers and the propagation of information between them. Secondly, gene-based approaches do not distinguish between reactive, independent, and causative relationships, whereas molecular components should be related causatively to the group of traits. For example, although transcript $g_2$ is reactive to the $p_4$ trait but does not cause it (*Figure 1—figure supplement 1A*), it is still part of a two-layer model (*Figure 1—figure supplement 1C*). Thus, a valid and reliable methodology for understanding similarities among distinct traits should identify a series of layers and causative relationships.

We have developed GEMOT, a methodology for constructing a causative model of trait–trait connections. GEMOT addresses the above challenges by constructing three-layer modules in which each module consists of a group of molecular mechanisms (here, gene transcripts within a 'transcripts layer') translating between a genetic layer and a layer of traits. In particular, a GEMOT module is focused specifically on causative, non-environmental relationships rather than on relationships of any kind, and accordingly it includes only transcripts that are part of causative relationships (referred to as 'driver transcripts', see *Figure 1A*, *Figure 1—figure supplement 1D*). Naively, systematic identification of GEMOT modules could be obtained by a detailed reconstruction of all relationships among variants, traits and transcripts (e.g., *Neto et al., 2010*; *Hageman et al., 2011*; *Wang and van Eeuwijk, 2014*). However, such a detailed reconstruction is a major computational and statistical burden, especially considering the large number of components. This problem is avoided in GEMOT by the use of a stepwise heuristic approach that drastically reduces the search space and allows scalability to large networks.

We applied GEMOT to a large dataset of 1738 traits spanning a broad spectrum of physiological, biochemical, clinical, and behavioral traits that were measured across the genotyped BXD mouse recombinant inbred strains (a cross between the C57BL/6J and DBA/2J strains [*Wang et al., 2003*]).

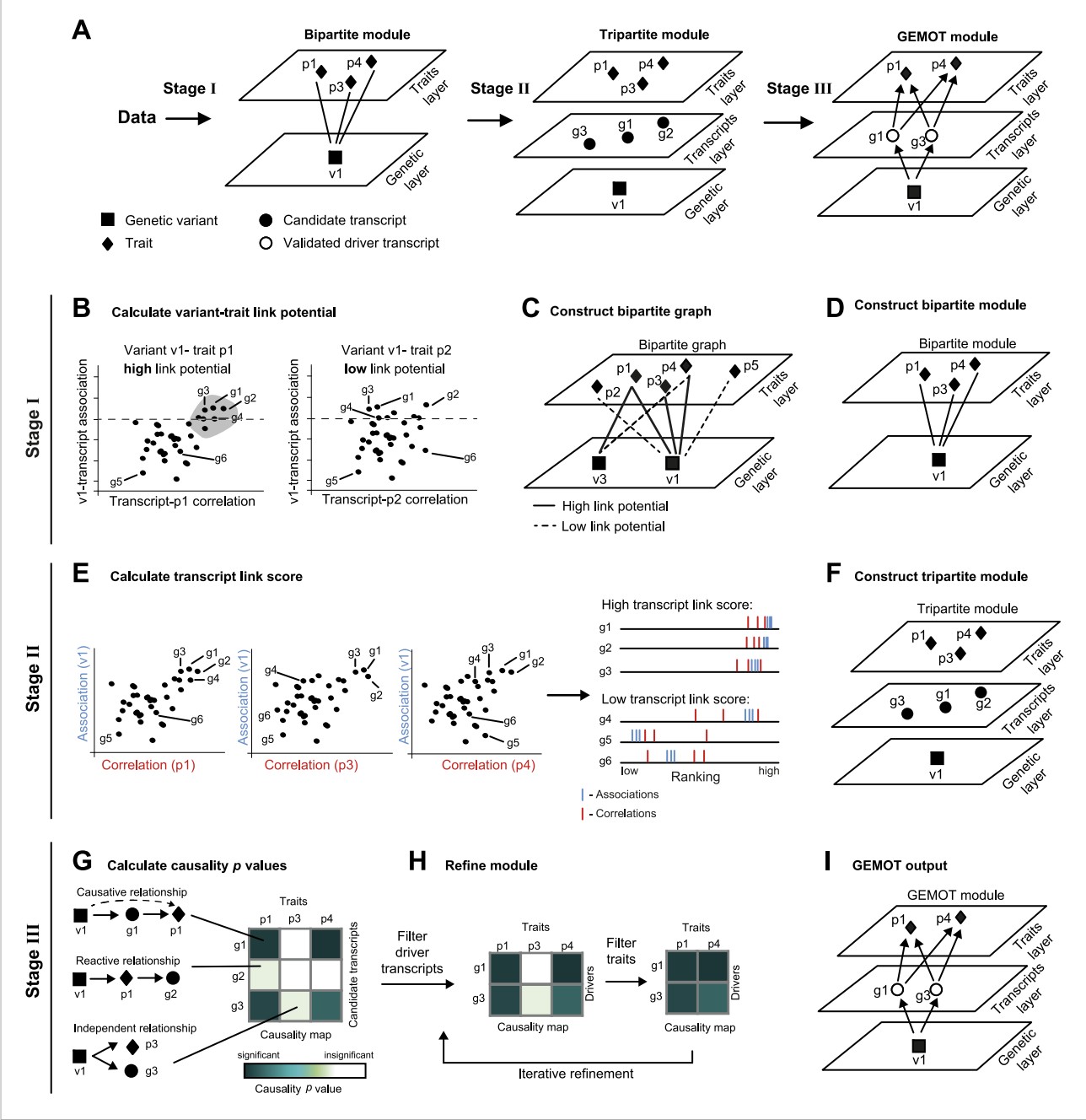

**Figure 1**. Schematic illustration of the GEMOT algorithm. (**A**) An overview of GEMOT for the scenario depicted in *Figure 1—figure supplement 1A*. GEMOT incorporates 3 stages (I, II and III) that are schematically described in **B–D**, **E–F** and **G–I**, respectively. Stage I: (**B**) High and low link-potential scores for pairs of variants and traits. A typical calculation for variant–trait pairs (left: $v_1$–$p_1$, right: $v_1$–$p_2$). Shown are variant–transcript associations (*y*-axis) and transcript–trait correlations (*x*-axis) for each transcript (black dots). GEMOT uses a threshold (horizontal dashed line) to compare the distribution of transcript–trait correlation in high and low transcript–variant associations scores. Link potential is high (left) when the distributions of correlation differ between the high and low association range more than expected by chance; link potential is low (right) where no difference is observed. Notably, a high link potential reflects the potential that some transcripts may translate between a variant and a trait, although such transcripts are not yet specified. (**C**) Bipartite graph construction. GEMOT constructs a graph whose two parts are variants (squares) and traits (diamonds); edge weights are the link-potential scores (solid lines, high; dashed lines, low). (**D**) Bipartite module identification. GEMOT identifies 'heavy' biclusters in the bipartite graph (in this example, 1 module). Stage II: (**E**) Transcript link score. The input is provided by all calculated correlation and association scores (such as the three plots on the left). On the right: given a transcript, GEMOT aggregates and ranks all its scores in a horizontal track (red, correlations; blue, associations) and uses the distribution of ranks to score the transcript for significance. (**F**) Tripartite module construction. GEMOT adds high-scoring transcripts from **E**, referred to as 'candidate transcripts' (circles), to the module. Stage III: (**G**) Causality p value scores. GEMOT uses a statistical score to assess causative relationships

*Figure 1. continued on next page*

*Figure 1. Continued*

(blue, significant; white, non-significant) for each transcript (row) and trait (column) in a module. Non-causative relationships attain non-significant scores (cartoon examples on the left). (**H**) Module refinement. Starting with the causality p value scores for the tripartite module (from **G**), GEMOT eliminates non-causative transcripts (left) and non-affected traits (right) in an iterative manner. (**I**) The resulting GEMOT module (arrows, causative relationships).

The following figure supplements are available for figure 1:

**Figure supplement 1**. An example of GEMOT methodology compared to alternative methods.

**Figure supplement 2**. Evaluation of methods for identifying (broad sense) causative relationships.

We used measurements of transcript levels in bone marrow-derived myeloid cells (*Gerrits et al., 2009*). The modules were used to uncover shared driver transcripts underlying collections of closely related or distinct traits. Notably, many of the findings were supported by independent knowledge or data. We also demonstrated the tissue specificity of modules, based on a post-processing analysis of gene expression in additional tissues. Our study highlights the power of causative reconstruction combined with grouping of complex traits to reveal a comprehensive picture of phenome connections.

## Results

### The GEMOT algorithm

Global identification of traits that share common causal transcripts and genetic mechanisms requires a reliable reconstruction of a global causal network among variants, transcripts and traits—a notoriously difficult problem, particularly in the case of a large collection of traits and high throughput data. We designed GEMOT, a scalable algorithm for the systematic construction of three-layer modules, each consisting of a group of traits, their shared causal driver transcripts and a genetic variant. GEMOT is based on the notion of 'linked relationships' between a variant, a transcript, and a trait. Such relationships incorporate a transcript that is both associated with a variant and correlated with a trait, without a direct evaluation of the causative relationship between the three components. In particular, it relies on the observation that causative relationships are expected also to be linked relationships (but not vice versa). It is therefore possible to start by constructing candidate modules based on the potential of variants and traits to be linked through transcripts. The internal structure is then constructed within each of these modules. Notably, the linked relationships are exploited to avoid global reconstruction of the particular types of relationships, which are then confirmed only at the validation stage.

GEMOT's input is a collection of traits, genotyping, and molecular data across a population of individuals. GEMOT consists of three main stages (see 'Materials and methods', *Figure 1A*). In stage I, GEMOT identifies 'bipartite modules' consisting of a single genomic interval and multiple traits that are connected by linked relationships through certain transcripts (e.g., traits $p_1$, $p_3$, $p_4$ and variant $v_1$ in *Figure 1A*, left). In stage II, candidate transcripts are added to the modules solely on the basis of their linked relationships (e.g., candidate transcripts $g_1$–$g_3$, *Figure 1A*, middle); the resulting 'tripartite modules', however, are not limited to causative relationships. Finally, in stage III, GEMOT refines the tripartite modules by investigating the causal relationships within them and eliminating non-causative transcripts. The output GEMOT module, therefore, finally consists of the validated driver transcripts, the trait(s) they affect, and a single causal genomic interval (e.g., driver transcripts $g_1$, $g_3$; traits $p_1$, $p_4$; variant $v_1$ in *Figure 1A*, right). Overall, each of the first two stages is aimed at filtering relevant objects for the next stage: candidate modules are selected on the basis of their potential to include candidate transcripts (stage I), and candidate transcripts are selected on the basis of an efficient score that is expected to be high in true driver transcripts (stage II). The final stage (stage III) is aimed at validating the causative relationships in the candidate modules from the previous stage. In the following we provide additional details about the three GEMOT stages.

 I. We start by calculating associations between each transcript and each variant (a 'variant–transcript association') and correlations between each transcript and each trait (a 'transcript–trait correlation'). We combine these two measures in a statistical test to identify variant–trait pairs that have high

potential to be linked through many transcripts. We call this scoring scheme a 'link potential' (*Figure 1B*). From these link-potential scores we construct a bipartite graph whose two parts are variants and traits, where edge weights are the link-potential scores (*Figure 1C*). We use a biclustering approach (the REL algorithm; [*Gat-Viks et al., 2010*]) to identify within this graph heavy 'bipartite modules', each consisting of a single genomic interval (harbouring a consecutive list of variants) and a collection of traits (*Figure 1D*). Importantly, such bipartite modules do not represent pleiotropy in general, but only pleiotropy that is likely mediated through transcripts.

II. We next apply a statistical test to identify transcripts whose linked relationships in the module are higher than expected by chance. This is done by evaluating the correlations and associations of transcripts with the module's traits and genomic interval respectively, computing 'transcript link scores', and using it to filter promising 'candidate transcripts' (*Figure 1E*). We then add the candidate transcripts to the bipartite modules, thus forming 'tripartite modules' (*Figure 1F*).

III. In the third stage the aim is to investigate the internal relationships within a module, thus allowing identification of driver transcripts and their affected traits. Recent methods allow structural reconstruction of a causality network (e.g., *Neto et al., 2010*; *Hageman et al., 2011*; *Wang and van Eeuwijk, 2014*), and can therefore be applied on each module to reveal its internal structure. However, since such methods may fail owing to a lack of scalability to large networks, we use an alternative approach that aims only to identify the relationships among layers, without the need to infer the causative relationships within each of the layers. To that end, we devise the following 2-step procedure: We first assess the causality among all candidate transcripts and traits in a module, assuming a single representative variant for the module's genomic interval (*Figure 1G*). We use a 'causality p value' score to assess the quality of such causative relationships. Next, the modules are refined by the iterative removal of transcripts and traits whose causality p values are non-significant (*Figure 1H*). The output is a list of 'GEMOT modules', each consisting of a single genomic interval, a group of validated 'driver transcripts', and their affected traits (*Figure 1I*).

Notably, the GEMOT algorithm is a unified pipeline that integrates several independent procedures, including biclustering, causality testing and network reconstruction. In this study we use the ReL biclustering algorithm (*Gat-Viks et al., 2010*); the causality testing proposed by *Neto et al. (2013)*; and a tailored network reconstruction scheme. However, the GEMOT pipeline is general and can be applied using alternative procedures (e.g., biclustering [*Tanay et al., 2002*]; network reconstruction [*Neto et al., 2010*; *Hageman et al., 2011*; *Wang and van Eeuwijk, 2014*]; causality testing [*Lee et al., 2009*; *Neto et al., 2013*]).

## Application of GEMOT in phenotypically diverse recombinant inbred mouse strains

We applied GEMOT to study phenome connections using gene expression in myeloid Gr-1+ cells (*Gerrits et al., 2009*) and 1738 traits across recombinant inbred BXD mice (see 'Materials and methods'). Using GEMOT, we found 40 bipartite modules, 11 tripartite modules, and 8 mature GEMOT modules with non-overlapping sets of drivers (permutation-based FDR < $10^{-3}$, 'Materials and methods'; see *Figure 2A*, *Figure 2—figure supplements 1, 2* and *Supplementary file 1A,B*). For comparison, no GEMOT modules were found in 100 permutation tests, and the average number of bipartite and tripartite modules in 100 permutation tests was 32 and 0.06, respectively (*Figure 2—figure supplement 2*). As expected, the observed link potential scores in mice were much higher than would be expected by chance (p < $10^{-10}$, permutation test; *Figure 2—figure supplement 3* and 'Materials and methods'). It is highly unlikely, therefore, that our GEMOT modules were generated at random (p < 0.01). Notably modules nos. 1–3, 6–8 show strong correlations between traits, whereas the remaining modules show moderate or weak trait–trait correlations (*Figure 2—figure supplement 4*).

Given that our findings were significant, we next aimed to obtain a global perspective on the resulting modules. To that end we generated a graph of the modules, where the transcripts layer of each module is connected to the module's traits and genomic interval (*Figure 2B*). On this graph we marked groups of similar traits, such as different behavioral and physiological responses after ethanol stimulation, or the results of different thermal nociception tests. Notably, the trait collections of the different modules are partly overlapping. For example, five of the modules relate to morphine responses (nos. 2, 5–8) and two modules relate to thermal nociception (nos. 3 and 5). Nevertheless, although some of the modules have overlapping traits, there is no overlap between the drivers and

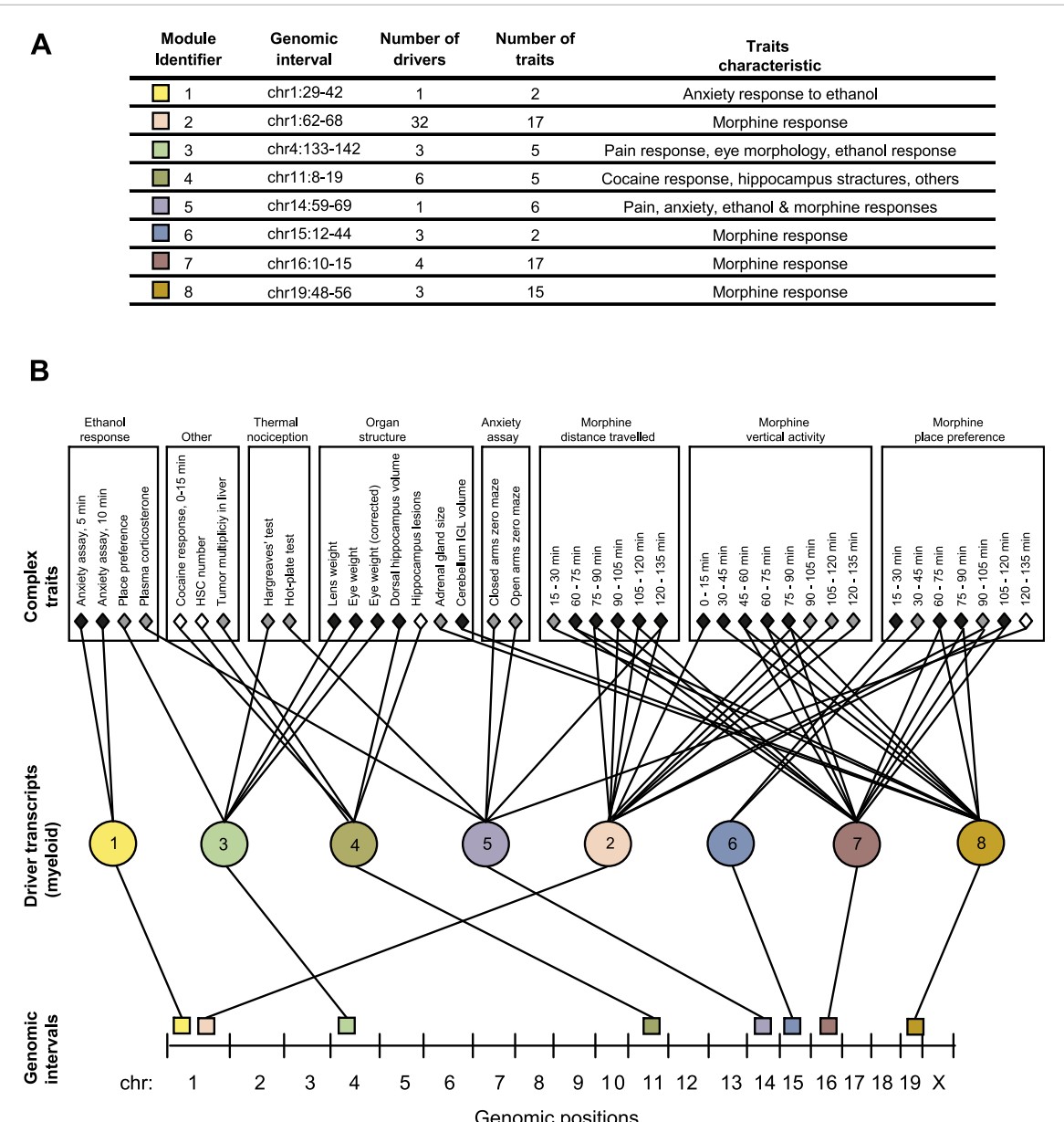

**Figure 2**. GEMOT modules in BXD mouse strains. (**A**) Shown is a module identifier (column 1), its genomic interval (column 2), the numbers of driver transcripts and traits in the module (columns 3 and 4, respectively), and the main characteristic of its traits (column 5, see description in **Supplementary file 1A**). (**B**) Global visualization of the GEMOT modules. The graph presents the genomic intervals (squares, bottom), the transcripts (circles, middle) and traits (diamonds, top) of all eight resulting GEMOT modules. The transcripts, which are unique to each module, are connected to their module's variants and traits. Sets of traits known to be related to the same process are enclosed in a box and labeled (top). The module's genetic and transcripts layers are color coded as in **A**; the traits are color coded based on their gender: female (white), male (gray) or both (black). Notably, some modules have overlapping collections of traits, or their traits relate to the same process.

The following figure supplements are available for figure 2:

**Figure supplement 1**. Linkage disequilibrium.

**Figure supplement 2**. Application of GEMOT on real and on permuted data.

**Figure supplement 3**. Higher link potential scores for real data than for permuted data.

*Figure 2. continued on next page*

*Figure 2. Continued*

**Figure supplement 4**. Trait- trait correlations in GEMOT modules.

**Figure supplement 5**. Characterization of GEMOT modules in BXD mouse strains.

variants of the different modules (*Figure 2A* and *Supplementary file 1B*), demonstrating GEMOT's ability to predict several distinct underlying mechanisms for the same collection of traits. Furthermore, a focused examination revealed that the traits of each of the modules shared a unique characteristic. For example, module nos. 6 and 2 relate to place preference following morphine injection, but at distinct time intervals (*Figure 2*); similarly, module nos. 1 and 3 relate to an anxiety assay, but with and without ethanol stimulation, respectively. Taken together, our results indicated high-level organization of overlapping collections of traits, while each module reflects a unique molecular and genetic signature that underlies a different trait characteristic.

Notably, some of the resulting modules consisted of multiple traits that are related to the same process, whereas others consisted of a collection of distinct traits (*Figure 2* and *Supplementary file 1B*). For example, a module related to morphine response (module no. 2) consists of 17 different traits that were measured following treatment of mice with morphine at different time points and in various behavioral assays. Similarly, a module related to an anxiety assay (module no. 1) consists of two different traits that were measured following treatment with ethanol at different time points. In contrast, and consistently with our goal of identifying novel relationships among traits, module nos. 3, 4 and 5 suggest previously unknown connections between traits.

We next characterized pairs of traits within each group of traits ('trait pairs') to show that the quality of these pairs is not lower than in existing methods. We focused on three main properties of trait pairs: the correlation among traits in a pair; the correlation between a trait pair and the transcripts; and the knowledge-based relationships among traits. As a reference we demonstrate these properties for modules that were generated using three alternative methods: (i) the trait-based hierarchical clustering approach (denoted 'HC'; [*Hastie et al., 2009*], as in *Figure 1—figure supplement 1B*); (ii) the gene-based INVAMOD algorithm, which identify pleiotropic genetic variants and their associated groups of traits in an agglomerative manner (*Gat-Viks et al., 2013*; as illustrated in *Figure 1—figure supplement 1C*); and (iii) a set of randomly sampled trait pairs (5% of all possible trait pairs). In particular, the bipartite modules were compared to the top 40 groups from each method (*Figure 2—figure supplement 5A,B*). Similar results were obtained when the eight GEMOT modules were compared to the top eight groups from each of the alternative methods (*Figure 2—figure supplement 5C,D*). For each property, we first explain the metric of evaluation and then present the results with GEMOT and with the alternative methods.

## Correlation among traits

GEMOT's correlations among trait pairs were much stronger than expected by chance ($p < 10^{-200}$), and were comparable to those obtained by the gene-based INVAMOD approach but weaker than those from the trait-based HC approach ($p < 10^{-160}$; *Figure 2—figure supplement 5E*). This result was consistent with the fact that HC, but not GEMOT or INVAMOD, directly optimizes for such correlations.

## Shared transcripts

For each trait pair we searched for a potential shared transcript that showed the best correlation (on average) with both traits. We found that GEMOT's predicted trait pairs were supported by the best-correlated transcripts across a wide range of trait–trait correlations (*Figure 2—figure supplement 5A,C*). As expected, in all methods the higher the correlation between traits, the stronger the correlations with the shared transcripts. This analysis indicated that GEMOT outperforms the gene-based and random methods ($p < 10^{-30}$ and $p < 10^{-50}$, respectively) and is comparable to the trait-based method.

## Knowledge-based relations among traits

We next aimed at determining whether trait pairs predicted by GEMOT were supported by previously known connections between traits. Evaluation of such distinct trait pairs in light of previously known

trait connections is a general problem, and no suitable systematic annotation is currently available. To tackle this problem we constructed an unbiased matched-annotation set of connections, and used it in our analysis as the gold standard. To systematically cover, independently of the correlation and co-association measures, the qualitative knowledge about connections among traits, we adopted the descriptive title of each trait in the GeneNetwork Phenotype Database (*Wang et al., 2003*). If the descriptions of two traits included at least one shared biological term, the traits were considered as 'matched-term traits' and were included in our gold standard set. Using this matched-annotation set, we found that all methods attained a similar proportion of matched-term trait; the higher the correlation between traits—the higher the proportion of matched terms, as expected (*Figure 2—figure supplement 5B,D*).

In the following we demonstrate three reconstructed modules, demonstrating GEMOT's ability to identify a model for a collection of tightly related traits, and for a previously uncharacterized combination of traits.

## Discovery of a model for a collection of tightly related traits

The morphine module (module no. 2, see *Figure 3*) exemplifies the ability of GEMOT to suggest an underlying mechanism for an entire group of traits known to be tightly related. The module consisted of a collection of 17 behavioral assays in the recombinant mouse strains, all carried out to measure their responses to injection of morphine (50 mg/kg, over different periods of time). Measured parameters included distance traveled, place preference, and vertical activity ([*Philip et al., 2010*]; *Supplementary file 1A*). All module traits showed a strong positive correlation with each other (|r| values ranged from 0.6 to 0.99, *Figure 2—figure supplement 4*) and shared similar peaks within the genomic interval of module no. 2 (*Figure 3A*, top).

Module no. 2 consists of a group of 32 driver transcripts (e.g., *Klf7*, *p35*, *Lrrk2*), all associated with the module's genotype and strongly correlated with each other and with the module traits (*Figure 3A*, bottom; *Figure 3B*; *Figure 3—figure supplement 1A*). For all driver transcripts, the causative relationships were much preferable to the alternative relationships (p value $\leq$ 0.005, permutation-based FDR $< 6 \times 10^{-5}$, 'Materials and methods'; *Figure 3C* and *Figure 3—figure supplement 1B*). We found two main groups of drivers (*Figure 3D,E*, *Figure 3—figure supplement 1A*). The first (denoted 'group I') consists of 21 genes whose transcript levels are negatively correlated with the morphine traits. In this group, individuals carrying the DBA/2J allele in the module's variant have higher gene expression values than those of individuals carrying the C57BL/6J allele (e.g., *Klf7* in *Figure 3—figure supplement 2A*). The second driver group (denoted 'group II') has the opposite correlation with the morphine traits and the opposite genetic effect (e.g., *Idh1*, in *Figure 3—figure supplement 2B*). These observations coincide with the fact that for all module traits, individuals carrying the C57BL/6J-allele have higher trait values (*Figure 3—figure supplement 2C,D*). Notably, causality p values in group I are more significant than in group II (p $<$ 0.05, t-test; *Figure 3C*); one possible reason is that the two groups might relate to distinct mechanisms that differ in their causality strength.

The role of the *Klf7* gene in morphine module no. 2 is particularly interesting. *Klf7* and *Idh1* are the only two *cis*-associated module drivers and their causative role in mediating morphine traits is highly significant. We hypothesized that the *cis*-associated variation in gene-expression levels may lead to variation in the *trans*-associated module drivers. To test this hypothesis we used the causality p value score ('Materials and methods'), but utilized a *cis*-associated gene (*Klf7* or *Idh1*) as the transcript positioned between the module's genomic interval and another module transcript. Using these scores we found that the causative p values of *Klf7* on the remaining module transcripts were substantially more significant than the causative p values of *Idh1* on those transcripts (*Figure 3F*, *Figure 3—figure supplement 3A*, p $< 10^{-7}$, K-S test). This finding holds for each of the driver groups I and II (*Figure 3—figure supplement 3B*), suggesting that *Klf7*, but not *Idh1*, likely affects the other module drivers, which in turn affect behavioral activity in response to morphine.

The finding of positive and negative correlations of *Klf7* with the drivers in groups I and II is particularly intriguing because it suggests that Klf7 is an activator of group I and a repressor of group II (*Figure 3D*). To validate the suggested central role of *Klf7* in mediating variation in other drivers, we analyzed the influences of knockdown and overexpression of *Klf7* on gene expression in bone-marrow hematopoietic stem cells from the C57BL/6J mouse strain (termed *Klf7*$^{KD}$ and *Klf7*$^{OE}$, using three and four biological repeats, respectively; data were taken from [*Schuettpelz et al., 2012*]). Indeed,

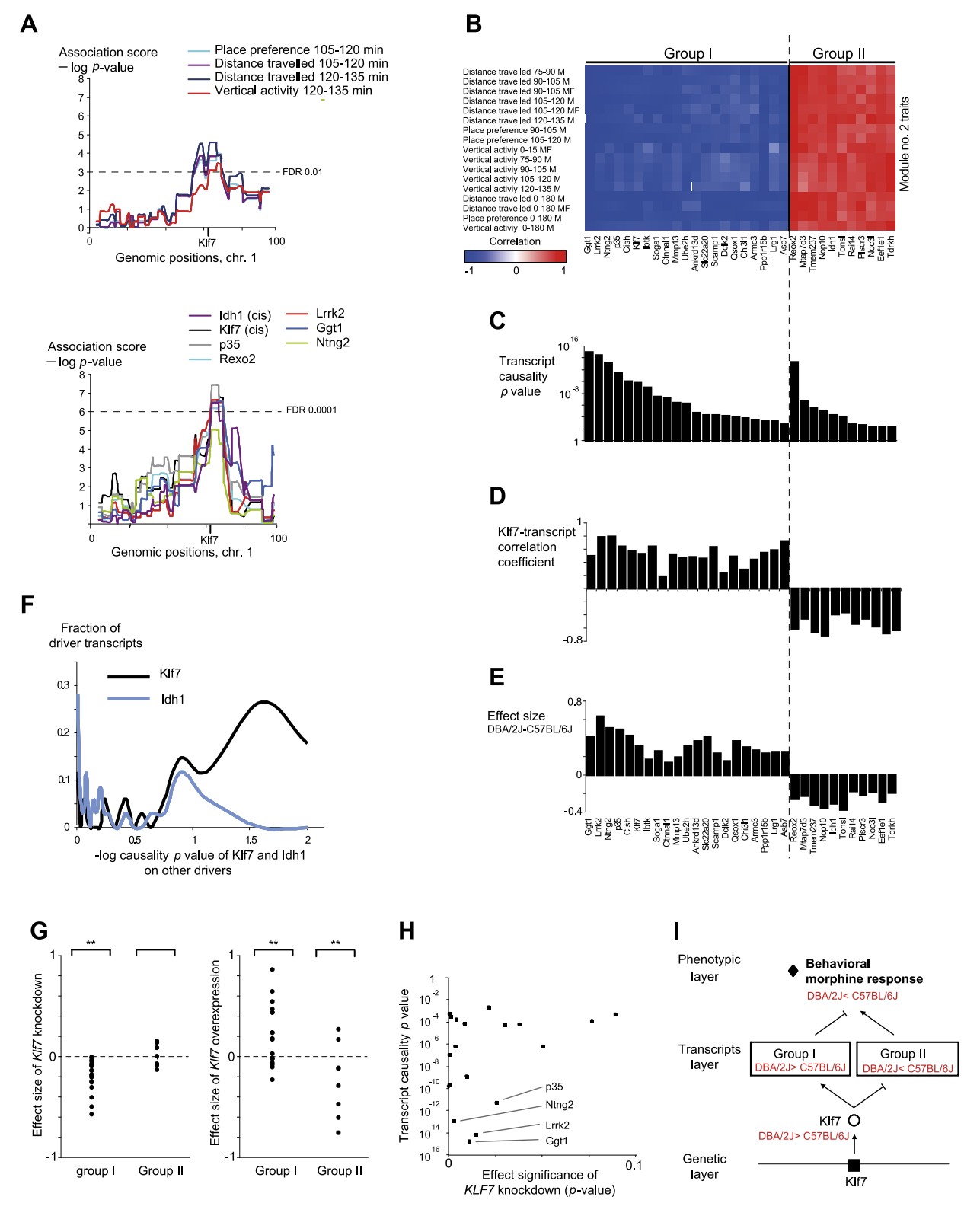

**Figure 3**. Characteristics of module no. 2. (**A**) Genetic associations. Shown are the association scores (*y*-axis) across the genomic positions of chromosome 1 (*x*-axis) for four module traits (top) and for seven selected drivers (bottom) in myeloid cells. The position of the *Klf7* gene is marked below the *x*-axis. (**B–E**) Characterization of driver groups I and II. (**B**) A matrix of traits (rows) vs drivers (columns), where the blue/red scale indicates negative/positive Pearson correlation coefficients among them. (**C**) The transcript causality p value scores (*y*-axis) are shown for each module transcript (log scale), assuming

**Figure 3. Continued**

a representative variant in the module's genomic interval (rs13475891 in chr1:62 Mbp). (**D**) The histogram represents the Pearson correlation coefficient (y-axis) of *Klf7* with the remaining drivers (x-axis). (**E**) Genetic effect size of variant rs13475891 on the driver transcripts. The histogram represents the average expression levels of DBA2-carrying individuals minus the B6-carrying individuals (y-axis) for each module driver (x-axis). (**F**) Distribution of the causality −log p value scores of *Idh1* (blue) and of *Klf7* (black) on each of the remaining drivers in the module. Causality p values were calculated by positioning *Idh1* or *Klf7* between the module variant and a driver from this module. (**G**) Validation of the effect size of *Klf7* perturbation of knockdown (left) or overexpression (right) on other drivers in bone-marrow hematopoietic stem cells (y-axis). The 'effect size' of perturbation (either knockdown or overexpression) on a certain transcript *g* is defined as the difference between the (log-scaled) expression of *g* in normal cells to that in the perturbed cells. In each panel, the first and second columns refer to groups I and II, respectively. **a significant t-test p value for determining whether the mean effect size is different from zero (FDR < 0.06). (**H**) Scatter plot, where for each driver in group I (a black square) the y-axis shows the transcript causality score (for morphine-response traits in this module; p values), and the x-axis shows the significance of *Klf7* knockdown effect on its transcription level (t-test p value). The plot indicates that in group I, drivers with highly significant causality on behavioral responses to morphine were also significantly influenced by *Klf7* knockdown. (**I**) Overall model illustration of module no. 2.

The following figure supplements are available for figure 3:

**Figure supplement 1**. Characterization of module no. 2 drivers.

**Figure supplement 2**. Relationships among components in module no. 2.

**Figure supplement 3**. Causality of *cis*-associated transcripts.

**Figure supplement 4**. Causative relationships of module no. 2 across multiple tissues.

**Figure supplement 5**. Causative relationships of module no. 2 across time points.

whereas knockdown of *Klf7* led to a down-regulation of group I transcripts ($p < 1.5 \times 10^{-5}$, FDR $< 5 \times 10^{-5}$, t-test), it had no influence in group II ($p > 0.32$; ***Figure 3G***, left), in agreement with the predicted role of *Klf7* as an activator and repressor of groups I and II, respectively. Furthermore, we found that the more significant the influence of *Klf7* knockdown on a driver in group I, the stronger the causative role of the driver on behavioral morphine traits (***Figure 3H***). Overexpression of Klf7 had the opposite effect, with significant down-regulation of transcripts in group II ($p < 0.05$, FDR $< 0.06$, t-test) and even a small increase of transcripts in group I ($p < 0.002$, FDR $< 0.005$; ***Figure 3G***, right), consistently with our model. Taken together, the two lines of evidence—both natural and experimental perturbations—indicated that *Klf7* is a key driver mediating the effects of additional drivers in groups I and II, which in turn affect morphine response diversity (***Figure 3I***).

Module no. 2 may affect morphine responses through a variety of mechanisms. For example, the *p35/Cdk5* driver directly phosphorylates the opioid receptor (***Xie et al., 2009***; ***Pareek et al., 2012***), and the morphine adduct MO-GSH is controlled by the *Idh1* and *Ggt1* drivers (***Correia et al., 1984***; ***Kumagai et al., 1990***; ***Muller and Do, 2012***). Furthermore, morphine treatment may exert its action through cell migration and cell invasion processes (***Gach et al., 2011***): the *p35* driver, as part of the *p35/Cdk5* complex, affects the Rac/Cdc42 complex through PAK inhibition, resulting in altered cell migration (***Nikolic et al., 1998***), while the *Mmp13* driver alters cell migration in response to morphine treatment because of its ability to degrade collagen (***Gach et al., 2011***; ***Wang et al., 2013***). Both the *Klf7* and the *Cdk5/p35* drivers activate p27 by expression or phosphorylation (***Laub et al., 2001***; ***Smaldone et al., 2004***), and p27 in turn affects the Rho GTPases Rac and RhoA, which then alter cell invasiveness and infiltration (***Kawauchi et al., 2006***). In glioblastoma, for example, the p27/Rho pathway affects infiltration of tumor cells (***Ruiz-Ontañon et al., 2013***). In agreement with this prediction, using the Rembrandt database (***Madhavan et al., 2009***) we found that all four top-ranked *Klf7*-mediated drivers attain significant effects on survival of patients with glioblastoma (Kaplan–Meier $p < 1 \times 10^{-11}$, $1.4 \times 10^{-4}$, $1 \times 10^{-4}$ and $0.01$ for the four drivers *Ntng2*, *p35*, *Rexo2* and *Lrrk2*, respectively; in all cases, FDR $< 0.05$, ***Figure 3—figure supplement 1C***), supporting the role of module no. 2 in cell invasiveness. Further experimental studies are required in order to test these suggested pathways and search for additional mechanisms.

Our findings in module no. 2 agree well with previous studies showing that several driver genes participate in the morphine response. For example, the *Klf7* transcript is up-regulated in response to morphine (being one of the top ten up-regulated genes [*Suzuki et al., 2003*]). Both the p35 driver and its activated protein Cdk5 were up-regulated in response to acute morphine but down-regulated on exposure to chronic morphine (*Ferrer-Alcón et al., 2003*). The coregulated *Idh1* and *Ggt1* drivers (*Muller and Do, 2012*) are responsible for the synthesis and degradation, respectively, of the reduced form of glutathione (GSH). A conjugated form of morphine and GSH (MO-GSH) attains higher morphine reactivity (*Correia et al., 1984*; *Kumagai et al., 1990*) that may alter the morphine responses. However, whereas key roles of *Klf7* have been reported primarily in neurons (*Bieker, 2001*; *Laub et al., 2005*), here we found a causative effect of *Klf7* on behavioral responses to morphine that was specific to myeloid tissue (*Figure 3—figure supplement 4*). We cannot as yet explain this observation; however, p35/Cdk5-mediated neutrophil secretion (*Rosales et al., 2004*) and the cytokine-mediated regulation of *Klf7* by morphine in lymphocytes (*Suzuki et al., 2003*) potentially provide an explanation for this tissue specificity. Furthermore, morphine injection leads to a reduction in neutrophil infiltration 30–120 min after treatment (*Clark et al., 2007*), in agreement with the timing of causative relationships between *Klf7* and behavioral assays at 30–120 min after morphine injection (*Figure 3—figure supplement 5*). Taken together, our results suggest that in vivo behavioral responses to morphine are affected not only by neuronal activity, but also through certain components of the immune system.

## Discovery of novel connections among traits

In the following we demonstrate GEMOT's ability to identify a model for previously uncharacterized connections, with either strong (module no. 3) or weak (module no. 4) correlations among traits.

Module no. 3 (*Figure 4A*) shows the ability of GEMOT to group a variety of distinct traits. This module consists of a genomic interval in chr4:133–142 Mbp and five correlated traits, namely pain response (thermal nociception [*Philip et al., 2010*]), lens weight (*Zhou and Williams, 1999*), eye weight (with or without correction for brain weight [*Zhou and Williams, 1999*]), and ethanol response (place preference [*Cunningham, 1995*]). All traits were found to be strongly intercorrelated, with |r| values ranging from 0.67 to 0.9 (*Figure 2—figure supplement 4*). *Eya3* and *Cd52* were proposed as *cis*-associated drivers, a suggestion further supported by the known role of *Eya3* in eye development (*Tadjuidje and Hegde, 2013*) and the involvement of *Cd52* in pain signaling (*Poh et al., 2012*). *Figure 4—figure supplement 1* shows that the significant causative role of these drivers can be found along the entire myeloid pathway (including stem cells [Lin⁻Sca-1⁺c-Kit⁺], common progenitors of the myeloid and erythroid lineages [Lin⁻Sca-1⁻c-Kit⁺], erythroid [TER-119⁺] and myeloid [Gr-1⁺] lineages), but not when using data from the lymphoid, eye, or brain tissues. This suggests that *Eya3* and *Cd52* play a role in pain processes and eye conditions mainly through their functionality in myeloid cells.

Module no. 4 (*Figure 4B*) demonstrates GEMOT's ability to identify a group of traits that show weak correlations among themselves, but share the same driver transcripts. The module consists of a genomic interval in chr11:8–19 Mbp and five traits: the measures of two hippocampal structures (volume and age-related lesions) from two distinct publications (*Jucker et al., 2000*; *Martin et al., 2006*), locomotor response to cocaine (*Philip et al., 2010*), number of liver tumors (*Lee et al., 1995*), and number of haematopoietic stem cells (*Liang et al., 2007*). The module's traits show weak and moderate correlations, with |r| values ranging from 0.01 to 0.58 (*Figure 2—figure supplement 4*). Of six suggested drivers, *Ythdf2* and *Aldh6a1* were predicted as the top drivers (*Figure 4B*, right panel). The validity of this prediction is supported by the known involvement of *Aldh6a1* in brain structure (*Marcadier et al., 2013*), and is further supported by a recent report (*Meyer and Jaffrey, 2014*) that N6-methyladenosine (m6A) modification of RNA, whose readers are *Ythdf1–3*, causes an altered locomotor response to cocaine. In this module, a significant causative role was found in myeloid cells, but not in brain, eye, liver, or lymphoid tissue (*Figure 4—figure supplement 2*), suggesting a novel function of myeloid cells in regulating neurobiological and behavioral traits—an intriguing possibility that warrants future investigation.

## Systematic evaluation of the GEMOT algorithm using synthetic data analysis

We next investigated the ability of GEMOT to identify the subsets of traits that are caused by the same transcripts. We first focused on studying GEMOT's utility for small sub-networks of co-mapped

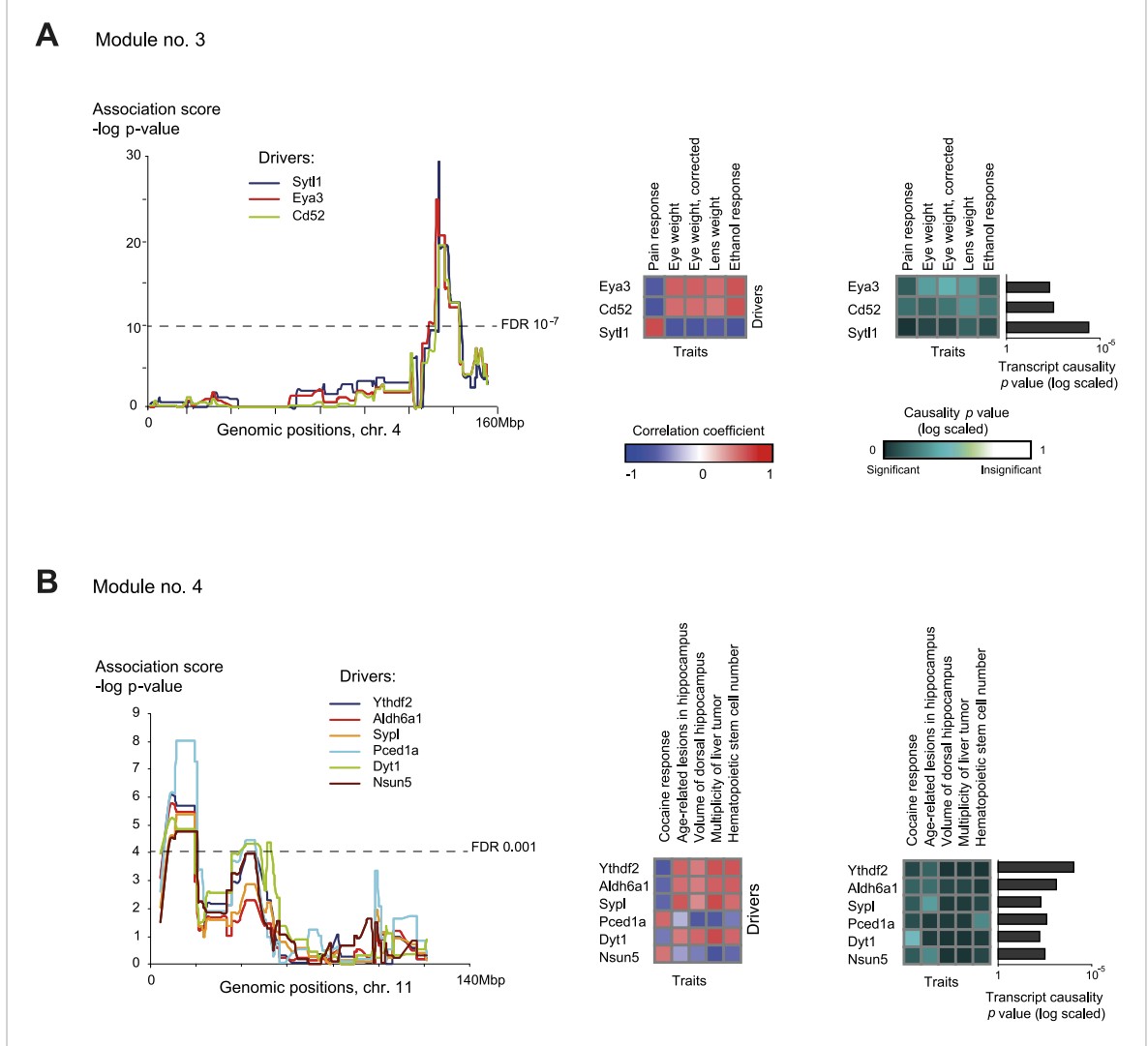

**Figure 4**. Characteristics of module nos. 3 (**A**) and 4 (**B**). Left: Genetic associations. Association scores (*y*-axis) across the genomic positions (*x*-axis) for the module's driver transcripts, based on gene-expression data in myeloid cells. Middle: Matrix of correlations among traits (columns) vs driver transcripts (rows); the blue/red scale indicates negative/positive correlation coefficients. Right: Matrix of traits (columns) vs driver transcripts (rows), where the blue/white scale indicates the significance of their causative relationships based on gene-expression data from the myeloid tissue. A histogram depicting transcript causality p value scores is shown as in **Figure 3C**. A representative variant in the module's genomic interval is assumed (see **Supplementary file 1C**).

The following figure supplements are available for figure 4:

**Figure supplement 1**. Causative relationships of module no. 3 in multiple tissues.

**Figure supplement 2**. Causative relationships of module no. 4 in multiple tissues.

components, where each sub-network consists of a subset of traits caused by the same transcripts, as well as additional transcripts and traits that are independently or reactively related (**Figure 5A**). Such sub-networks mimic the tripartite modules that serve as input at the third stage of the GEMOT algorithm. A single synthetic data collection consisted of genotyping, phenotyping, and gene expression for 100 such sub-networks with two characteristic parameters: number of traits and noise level; in all cases we used 100 individuals ('Materials and methods'). Using these synthetic data, GEMOT performance was compared to that of three alternative network reconstruction methods:

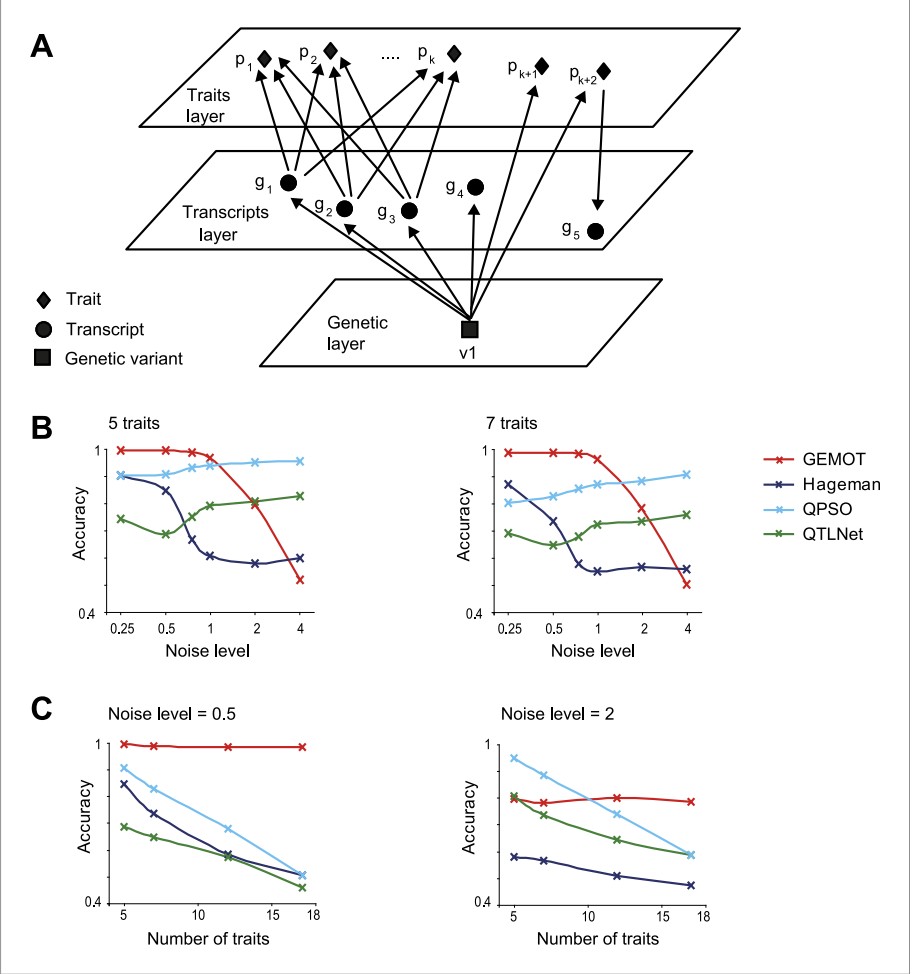

**Figure 5**. Comparative performance analysis on simulated models. (**A**) Illustration of a sub-network model, in which all components are mapped to the same genetic variant $v_1$ but not necessarily through the same relationships. In particular, the model includes $k + 2$ traits ,with $k$ traits $p_1,..,p_k$ that share the same underlying transcripts $g_1$, $g_2$, $g_3$, and two additional traits $p_{k + 1}$ and $p_{k + 2}$ that are affected through other mechanisms (see detailed in *Figure 5—figure supplement 1A*). (**B** and **C**) Accuracy assessments of the synthetic sub-network depicted in **A**. Accuracy (y-axes) is compared across methods and different data parameters. Results are shown in models of different noise levels (x-axis, log-scaled; **B**) with either 5 (left) or 7 (right) traits, or over different numbers of traits (x-axis; **C**) with either a low noise level = 0.5 (left) or a high noise level = 2 (right). The accuracy metric evaluates whether $p_1,..,p_k$, but not $p_{k + 1},..,p_{k + 2}$, share the same mechanisms, as detailed in *Figure 5—figure supplement 1B*. Plots depict alternative network construction methods (color coded, see *Figure 5—figure supplement 2*), indicating that GEMOT has an advantage over existing methods with noise levels ranging between 0.25 and 1, which is the relevant range for the mouse data in this study (see *Figure 5—figure supplement 3*).

The following figure supplements are available for figure 5:

**Figure supplement 1**. Accuracy assessment of predicted shared mechanisms in synthetic (co-mapped) sub-networks.

**Figure supplement 2**. Evaluation of the compared network reconstruction methods.

**Figure supplement 3**. Relevant range of synthetic data parameters.

**Figure supplement 4**. Accuracy assessment in additional sub-networks.

**Figure supplement 5**. Accuracy assessment in a large biological network.

*Hageman et al., 2011*; *Wang and van Eeuwijk, 2014* ('QPSO'); and *Neto et al., 2010* ('QTLNet') (see parameter selection in *Figure 5—figure supplement 2*).

Performance was evaluated using an accuracy metric, which reflects the ability of a method to discern the correct subset of traits sharing the same transcripts (e.g., traits $p_1,...,p_k$ but not $p_{k+1}$ and $p_{k+2}$ in *Figure 5A* and *Figure 5—figure supplement 1B* for details). *Figure 5B* presents the accuracy for synthetic data collections of varying levels of noise (using the sub-networks from *Figure 5A* with 5 or 7 traits). GEMOT displayed the best accuracy for noise levels ranging between 0.25 and 1, with lower accuracy for higher noise levels. Analysis of various network properties in both mouse and synthetic data shows that sub-networks with a noise level that do not exceed 1 are more likely to represent real biological modules (*Figure 5—figure supplement 3*). Unlike the compared methods, GEMOT's accuracy remained high with an increasing number of traits (*Figure 5C*); similar results were obtained for alternative network structures (*Figure 5—figure supplement 4*). Overall, in our simulation, GEMOT outperformed the compared algorithms in handling a growing number of traits and in identifying the correct groups of traits when using biologically-relevant parameters. These results do not rule out the possibility that for other tissues, conditions or organisms, utilizing the alternative methods as part of the third stage of the GEMOT algorithm may enhance its performance.

We next aimed to characterize GEMOT's utility for a large biological network that included groups of traits that share the same causal transcripts. Accordingly, each synthetic network included 100 traits, 200 transcripts and 100 variants, featuring five co-mapped sub-networks. A singe data collection consists of 100 networks, each containing five co-mapped sub-networks that carry the same number of traits ('Materials and methods'). We compared GEMOT to two alternative trait-grouping methods: the trait-based iterative clique enumeration (ICE) approach (*Shi et al., 2010*) and the gene-based INVAMOD approach (*Gat-Viks et al., 2013*) ('Materials and methods'). Notably, network construction methods (e.g., *Neto et al., 2010*; *Hageman et al., 2011*; *Wang and van Eeuwijk, 2014*) could not be compared owing to an unrealistic running time in the case of large networks. The analysis suggested that although all compared methods successfully discern all traits in a sub-network (*Figure 5—figure supplement 5A*), GEMOT attains higher accuracy in discerning those traits that share the same driver transcripts (*Figure 5—figure supplement 5B*). Notably, the GEMOT algorithm is tailored for identification of causative relationships, unlike the compared methods, explaining why GEMOT succeeded in discriminating the correct subsets of co-regulated traits.

## Discussion

We set out to identify the molecular and genetic mechanisms underlying connections among groups of traits. To that end, we combined module identification (*Gat-Viks et al., 2010*) with causality testing (*Neto et al., 2013*) in a unified pipeline that relies on the definition of linked relationships so that candidate modules can be filtered out prior to the validation stage. Our results in mice highlighted three types of high-order organization of traits. (i) Groups of tightly related traits that share the same transcripts mechanisms (modules 1, 2, 6, 7, 8, e.g., *Figure 3*). (ii) Groups of distinct traits that share the same transcripts mechanism, but not necessarily high correlations among them (modules 3, 4, 5, e.g., *Figure 4*). (iii) Different groups commonly have overlapping traits, but typically differ in their underlying mechanisms (*Figure 2B*).

Our study emphasizes the need for methodologies for constructing causative models that underlie connections among traits. Whereas previous trait-grouping methods have used genetic or molecular data separately, and thus did not validate causative transcripts (e.g., *Shi et al., 2010*; *Gat-Viks et al., 2013*; as illustrated in *Figure 1—figure supplement 1B,C*), the GEMOT method aims at directly filtering and validating such relationships. Our simulations showed that GEMOT is superior to these methods in identifying trait groups that share the same underlying transcripts (*Figure 5—figure supplement 5B*). Another strategy is to use network reconstruction methods to construct groups of related traits (e.g., *Neto et al., 2010*; *Hageman et al., 2011*; *Wang and van Eeuwijk, 2014*). These methods can be applied in the case of either the complete biological network or the sub-networks within the tripartite modules. Whereas these methods are limited in their scalability and may be particularly inefficient when applied on a large number of components, our approach can be scaled to larger networks, but can construct the network only partially. For example, in comparing both running time and accuracy under increased network sizes we found GEMOT to be more scalable than the alternative network construction methods (*Figure 5C* and *Figure 5—figure supplements 2B, 4B*).

Our methodology opens the door to a variety of future research directions. One possibility is that GEMOT will be applied on a compendium of molecular data from multiple tissues. In such cases, GEMOT predictions can be used to simultaneously identify both the biological mechanism and its relevant tissue. Second, GEMOT is applicable to a variety of molecular data types in addition to gene-expression data. For example, its application in blood cytokines or plasma lipids (*Teslovich et al., 2010*) is expected to make it possible to identify molecular factors acting at the cell–cell communication level. Similarly, future extensions of GEMOT may provide the means to include environmental factors as part of the module. Third, monitoring of additional strains would allow discriminating between several alternative genomic intervals for the same module, which may arise due to linkage disequilibrium between chromosomally distinct loci. Fourth, characterization of GEMOT modules that share a similar collection of traits but have different genomic intervals may reflect gene–gene interactions that lead to connections among traits. Finally, GEMOT can potentially be further improved by the construction of internal causative relationships within the transcripts and the traits layers. For example, some drivers may control other drivers, which in turn affect a collection of traits (as exemplified in the case of *Klf7* in *Figure 3*). It should be noted, however, that GEMOT cannot distinguish cases of multiple drivers that are part of the same regulatory circuit from cases of multiple drivers that act through several distinct circuits. Rather, its predictions provide biological or clinical hypotheses for additional experimental investigations.

Overall, our approach paves the way to the simultaneous study of several mechanistic layers underlying connections among traits, providing a multilayered view of phenotypic connections. Because the GEMOT methodology is a general one and can be applied to the study of other taxa, this approach may facilitate our understanding of the molecular mechanisms underlying human disease.

## Materials and methods

### The GEMOT algorithm

GEMOT is designed to identify three-layer modules in which driver components translate between a single genomic interval and a collection of traits. As input, GEMOT takes three types of objects: (1) a collection of traits across a population of individuals; (2) genotyped genetic variants for these individuals; and (3) high-throughput gene expression data across the same population.

Our algorithm incorporates three stages (*Figure 1A*). The first stage constructs candidate bipartite modules consisting of a group of traits and a genomic interval. In the second stage, candidate transcripts are added to each module from the previous stage, thus forming tripartite modules. The final stage validates the actual drivers and refines the modules accordingly. The GEMOT code is available at http://csgi.tau.ac.il/gemot/.

### GEMOT stage I: construct bipartite modules

In the following we first define the construction of a bipartite graph and then explain the identification of bipartite clusters within this graph as previously described (*Gat-Viks et al., 2010*). We define a bipartite graph whose two parts correspond to genetic variants and traits, and in which the edges reflect the potential of a variant and a trait to have significant linked relationships (*Figure 1C*). Edge weights are calculated as follows (*Figure 1B*): First, for each pair of a genetic variant and a transcript, we evaluate the genetic association between the expression of the transcript and the candidate genetic variant. This yields a 'variant–transcript association score'. In this study, for the case of homozygous recombinant inbred strains the association score is a −log t-test p value for the different gene-expression values between the strains carrying the two possible variant alleles. For other cases, such as an outbred population, other standard association scores can be applied (*Falconer and Mackay, 1996*). Secondly, for each pair of a transcript and a trait, we calculate the absolute Pearson correlation coefficient across genetic backgrounds. We term this score the 'transcript–trait correlation score'. Finally, for each genetic variant and each trait we compare the distribution of transcript–trait correlations in high and low transcript–variant association scores (a statistical t-test). We assign such a t-test p value to five different transcript–variant association cutoffs (the five cutoffs partition the association range into 6 equally sized groups) and record the top −log t-test p value across these five cutoffs. We refer to the recorded −log p values as 'link potentials' and use them as edge weights in the bipartite graph.

Within this graph we use the ReL software package (*Gat-Viks et al., 2010*) to identify significant biclusters (*Figure 1D*). Briefly, the ReL algorithm starts with a set of seed clusters consisting of one trait and one variant whose link potential exceeds a certain initialization cutoff, $c_s$. A trait or a variant can be included in a cluster if and only if its average link potential exceeds an improvement cutoff $c_i$ (here, $c_s = 180$, $c_i = 90$). Each bipartite cluster is subject to iterative improvements by addition or removal of traits and variants based on this cutoff. We refer to the bipartite clusters as 'bipartite modules' and further improve them in the following stages. Notably, ReL provides the same results when applied with or without Boneferroni correction for the gene–variant association −log p values and link potentials scores, since the construction of the biclusters is robust to an additive rescaling of these scores.

## GEMOT stage II: construct tripartite modules

In this stage a rough list of candidate transcripts is constructed for each module (*Figure 1E*). To this end, for each transcript in a given module we rank its correlations and associations with all traits and variants in the input dataset and record its ranks of associations and correlations within the module. We next compare the distribution of recorded ranks in this module with the distribution of all ranks. The two distributions are compared using a Kolmogorov–Smirnov (K-S) test, a nonparametric test that may be used to compared two samples. We refer to the K-S p value for a transcript in a module as a 'transcript link score'. Only transcripts with significant link scores are added to their module. Such transcripts are called 'candidate transcripts' and the resulting extended modules are referred to as 'tripartite modules' (*Figure 1F*).

## GEMOT stage III: refine module (validate drivers)

In the following we first define the causality test and then describe the procedures for identifying driver transcripts and for module refinement.

### Causality testing

The input for a causality test is a triplet of objects—a variant, a transcript, and a trait—each measured across the same population of individuals. Several types of relationships exist, including causative, reactive, and independent, and the goal of causality testing is to identify causative relationships among a given triplet of objects. Our causality analysis here follows the formulation of *Neto et al. (2013)*. In this formulation, a likelihood score is first calculated for the causality model (denoted 'M1') as well as for three alternative models (denoted M2–M5, where M4 and M5 are modeled together, *Figure 1—figure supplement 2A*). A likelihood ratio score is then calculated between each of the three alternative models (as null hypotheses) and the causative model (as an alternative hypothesis), whereas the three respective p values are calculated on the basis of a theoretically derived distribution of the likelihood ratios. The final 'causality p value' is the maximal obtained for the three calculated p values.

Importantly, in the present study we introduce two important modifications in the scheme suggested by *Neto et al. (2013)*. First, the distribution of likelihood ratios was evaluated empirically by randomly reshuffling the transcript levels of the candidate transcript and recomputing the likelihood ratio (repeating the procedure 100 times). The causality p value of a given triplet of objects was calculated according to the distribution of these reshuffling-based likelihood ratio scores (rather than using a theoretically derived distribution).

Second, we observed that in a GEMOT module a transcript may have a causal effect on a trait even in the presence of additional transcripts through which the variant influences the trait. Accordingly, in the GEMOT framework, both models M1 and M4 should be considered as causative relationships (a 'broad' view of causality), as opposed to the classical definition of causative relationships according to model M1 only (*Figure 1—figure supplement 2A*). In the present study we therefore utilized the broad-sense causative relationships. We observed that samples of model M4 attained significant likelihood ratios of M1 against models M2 and M3, and we therefore defined the (broad sense) 'causality p value score' as the maximal across the p values attained for M1 against M2 or M3. Our simulation study (see 'Materials and methods') indicated that GEMOT attains similar performance to that of the compared methods for models M1–M3, but outperforms the existing methods when adding the remaining models (*Figure 1—figure supplement 2B–D*).

## Driver validation and module refinement

Transcripts that are causally related to traits can be identified by constructing a detailed causal network of relationships among all the components in the tripartite module. This, however, is a difficult task in the case of large tripartite modules harboring numerous components. We therefore devised an algorithm to identify the driver transcripts and their affected traits without needing to construct the module's entire network. The algorithm proceeds as follows: For each module, stage III starts with calculation of the causality scores for each pair of candidate transcripts and traits in the module, assuming a fixed single 'representative variant' from the module's genomic interval (*Figure 1G*). In this study, the representative variant was the one that attained the best average association scores across the module's candidate driver transcripts. Next, an iterative module refinement is applied in two steps. In step (i) we reveal the driver transcripts that are causally related to many of the module traits (*Figure 1H*, middle). In particular, we used Fisher's method (*Sokal and Rohlf, 1990*) to calculate the overall causality p value of a transcript (denoted 'transcript causality p value'); only those transcripts with significant transcript causality p values were retained. The transcript causality p value is determined based on chi-squared distribution with *2k* degrees of freedom: $\chi^2_{2k}(g_i) = -2 \sum_{j=1,...,k} \ln(p_{i,j})$, where $g_i$ is the transcript; $j = 1,...,k$ are the module traits; and $p_{ij}$ is the causality p value for the transcript $g_i$ and a trait $j$ assuming the module's representative variant. Next, in step (ii) those traits that do not have at least one transcript with a significant causality score are filtered out from among the transcripts that were selected in step (i) (*Figure 1H*, right). This process is repeated iteratively until convergence. Thereafter, the retained transcripts are called 'driver transcripts' and modules that contain at least one driver are termed 'GEMOT modules' (*Figure 1I*).

## Mouse data

All mice data was taken from a previously produced body of work. We applied our analysis to data obtained from homozygous BXD recombinant inbred mouse strains (*Peirce et al., 2004*) generated by crossing C57BL/6J and DBA/2J inbred strains for many generations. Microarray expression data in myeloid cells across 24 BXD strains have been measured (*Gerrits et al., 2009*). To identify high-quality candidates we selected 5786 genes whose variation in expression across BXD strains, based on average intensities of the genes, was higher than expected. Expected variance values were calculated using a sliding window along the genes' average intensities. Gene-expression values were $\log_{10}$-transformed and normalized by Z-score normalization. All 2885 traits and 3796 genetic variants across BXD strains were downloaded from the WebQTL dataset (*Wang et al., 2003*). Trait values were normalized by Z-score normalization. Given the strains in the gene expression and trait datasets, we restricted our analysis to 1738 traits that had at least 15 strains in common. Other compared cell types or tissues (*Gatti et al., 2007*; *Geisert et al., 2009*; *Gerrits et al., 2009*; *Alberts et al., 2011*; *Mozhui et al., 2012*) were similarly preprocessed. *Supplementary file 1C* records the particular representative variants that were used for causality testing in each predicted GEMOT module.

To assess the corresponding false discovery rates, we generated negative controls based on a permutation test in which the transcript levels of each transcript were randomly shuffled and the GEMOT modules were recomputed (a process that was repeated 100 times). In each repeat, a variety of statistics (such as the number of identified modules) were recorded. The permutation-based 'false discovery rate' (FDR) is the ratio of the averaged number of statistics that were declared significant using the permuted data to the number of statistics that were declared significant using the original (non-permuted) data. In this study, GEMOT was applied using transcript link score cutoff = $10^{-95}$ for identifying candidate transcripts (stage II, permutation-based FDR < 0.09) and transcript causality p value cutoff = 0.005 (stage III, permutation-based FDR < $6 \times 10^{-5}$).

## Causality testing–performance evaluation

To investigate the performance of the causality score we simulated triplets of objects, each consisting of a variant *v*, a transcript *g* and a trait *p*. In all such triplets we assume 100 homozygous individuals. The genotyping of each variant was generated by sampling a vector of values 0 and 1 from a binomial distribution (with p = 0.5). Based on these genotyping values, the values of *g* and *p* were generated according to the following five different models (denoted M1–M5), as depicted in *Figure 1—figure supplement 2A*: M1, a causative model v → g → p; M2, an independent model v → g, v → p; M3, a reactive model v → p → g; M4, v → g → p and v → p; and M5, v → p → g and v → g. Each arrow in

these five models was simulated as a linear expression with a normally distributed error term. For example, based on model M1, data were generated as $g = \alpha \cdot v + \varepsilon$, $p = \lambda \cdot g + \varepsilon$, $\varepsilon \sim N(0, \sigma^2)$; similarly, model M4 was generated as $g = \alpha \cdot v + \varepsilon$, $p = \alpha \cdot v + \lambda \cdot g + \varepsilon$, $\varepsilon \sim N(0, \sigma^2)$. A single 'synthetic collection' consisted of 250 relationships from each of the five models, a total of 1250 samples. Results are displayed for many different collections, each generated using different combination of $\lambda$ and $\sigma$ values with $\alpha = 0.5$. GEMOT's causality p values were compared to those obtained by two alternative methods: QTLHot (*Neto et al., 2013*) and an AIC-based method (*Lee et al., 2009*).

For a given significance threshold we evaluated the ability to identify causal relationships using true positive (TP), true negative (TN), false positive (FP) and false negative (FN) counts, which were defined according to the broad-sense definition of causality (*Figure 1—figure supplement 2B*). The area under the receiver operating characteristic (ROC) curve (the AUC) was calculated accordingly, where the higher the AUC the better the method. In addition, a balanced false discovery rate (FDR) can be used as a criterion for comparisons of methods, computed as $FDR = FP^a/(FP^a + TP)$ where $FP^a$ accounts for imbalanced data by dividing FP by the ratio between the negative and positive synthetic datasets, calculated as $FP^a = FP/\pi_0$, $\pi_0 = (FP + TN)/(TP + FN)$. The method with lowest FDR is regarded as the best method (when all methods use the same p value cutoff). Notably, GEMOT attains similar performance to that of the compared methods for models M1–M3 (*Figure 1—figure supplement 2C*), but outperforms the existing methods when adding synthetic M4 and M5 samples (*Figure 1—figure supplement 2D*).

## GEMOT performance analysis

The synthetic data analysis is focused on two simulations with increasing complexity of the input network: (i) sub-network analysis, in which the input is a sub-network where all components are co-mapped to the same variant; and (ii) network analysis, in which the input is a large network comprising several co-mapped sub-networks.

### Sub-network analysis

A synthetic co-mapped sub-network consists of one variant $v$ and $m$ transcripts $G^C = \{g_1^C, ..., g_m^C\}$ that are associated with the variant $v$ and are causally related to $k$ traits $P^C = \{p_1^C, ..., p_k^C\}$, that is, $v \rightarrow \{g_1^C, ..., g_m^C\} \rightarrow \{p_1^C, ..., p_k^C\}$. In addition, the sub-network includes a single transcript $g_1^R$ that is reactive to phenotype $p_1^R$ (thus, $v \rightarrow p_1^R \rightarrow g_1^R$) and a pair of a transcript $g_1^I$ and a trait $p_1^I$ that are independently affected by $v$ such that $v \rightarrow p_1^I$ and $v \rightarrow g_1^I$. Altogether, a sub-network consists of the triplet $(v, P, G)$, where $P = \{P^C, p_1^R, p_1^I\}$ and $G = \{G^C, g_1^R, g_1^I\}$. In this study we analyzed three sub-network structures, referred to as *Net-1* (*Figure 5* and *Figure 5—figure supplement 1A*), *Net-2* (*Figure 5—figure supplement 4A*, left), and *Net-3* (*Figure 5—figure supplement 4A*, right); unless stated otherwise, the results refer to Net-1. In all cases, the total 'number of traits' in a sub-network is $k + 2$. The simulation data was generated by using for each edge $a_i \rightarrow a_j$ in these sub-networks a linear coefficient $\beta$ such that $a_j = \beta \cdot a_i + \varepsilon(\beta)$, $\varepsilon(\beta) \sim N(0, \sigma^2)$, $\sigma = q_\varepsilon \cdot \beta$, where $q_\varepsilon$ represents the relative proportion of noise referred to as the 'noise level'. Observe that $\varepsilon(\beta)$ depends on both the linear coefficient of the relationship $\beta$ and on the noise level $q_\varepsilon$. We generated 'synthetic collections' of 100 sub-networks (100 individuals in each case); each collection is constructed for a given network structure (sub-networks Net-1, Net-2, Net-3), a certain total number of traits ($k + 2 = 5, 7, 12,$ or 17 traits) and a certain noise level ($q_\varepsilon = 0.25, 0.5, 1, 2, 4$). In all cases we used $\beta = 1.5$ for relationships $v \rightarrow p$ and $v \rightarrow g$ and $\beta = 0.6$ for relationships $g \rightarrow p$ and $p \rightarrow g$.

We compared GEMOT's performance on co-mapped sub-networks to that of alternative network construction algorithms, including the QTLNet method (*Neto et al., 2010*), the QPSO methodology (*Wang and van Eeuwijk, 2014*), and a Bayesian construction proposed by *Hageman et al. (2011)*; for all methods, we used the parameters that gave the best results for the tested networks (*Figure 5—figure supplement 2A*). We evaluated the performance of each method using the true positive (TP), true negative (TN), false positive (FP) and false negative (FN) counts as defined in *Figure 5—figure supplement 1B*, which depicts whether the network construction correctly identified shared transcripts for the traits in $P^C$. Relying on these counts, the balanced accuracy score was calculated as (TPR + TNR)/2 (here, TNR = TN/(TN + FP) and TPR = TP/(TP + FN)). AUC could not be computed because the three compared approaches do not provide a measure of statistical significance.

## Network analysis

The synthetic network consisted of two parts: first, five co-mapped sub-networks $s_1,...,s_5$ of the form $(v(s_L), P(s_L), G(s_L))$, and second, the remaining network comprising additional transcripts, variants, and traits. In total, the network consisted of 100 variants, 100 traits, and 200 transcripts that were used to generate synthetic data across 100 individuals. The data for the five sub-networks were generated as in the sub-network analysis ('Materials and methods'), using varying numbers of traits. The remaining data were generated as follows: (i) genotyping of each variant was generated independently, as described in the causality-testing simulation ('Materials and methods'); (ii) for each trait and individual, data values were sampled from a standard normal distribution; and (iii) gene-expression data were generated while maintaining the correlation between transcripts of the non-module components and between them and the module component. More specifically, for each individual $i$ and transcript $j$ that are not in a sub-network, the gene-expression data $z_{ij}$ were generated in two steps: first, to maintain correlation among all non-sub-network transcripts we sampled the real data in murine myeloid cells (data taken from *Gerrits et al., 2009*) to generate a $200 \times 200$ covariance matrix, and then used this matrix to generate synthetic data values $x_{ij}$ which approximately have the same covariance matrix. Next, to improve the correlation of the $x_{ij}$ values with at least one co-mapped sub-network, we calculated the gene-expression data as $z_{ij} = x_{ij} + c \cdot y_{ik}$, where $y_{ik}$ is the synthetic gene expression for an individual $i$ and the (arbitrarily selected) $k$th transcript in the sub-network (in all cases, $c = 1$).

The synthetic network was used to test the ability of GEMOT to identify the groups $P^C(s_u)$ (here, $u = 1,..,5$), which are the groups of traits with the same causative mechanism behind them. Furthermore, we tested an existing trait-based approach (Iterative Clique Enumeration (ICE); *Shi et al., 2010*) and a gene-based approach (INVAMOD, *Gat-Viks et al., 2013*). Two different measures were used for performance evaluation: first we evaluated grouping of traits based on their co-mapping to the same variant (*Figure 5—figure supplement 5A*). Secondly, we evaluated the ability of each method to identify groups of traits sharing the same causal transcripts (*Figure 5—figure supplement 5B*).

## Acknowledgements

We thank Prof. Marcelo Ehrlich for valuable discussions and comments, and Avital Brodt for artwork. This research is supported by the Israeli Centers of Research Excellence (I-CORE): Gene Regulation in Complex Human Disease, Center No. 41/11 (YO, RW); Israel Science Foundation (ISF) fund no. 1643/13 (YO, RW), ISF-Broad fund no. 1168/14 (AN, AF), and the Edmond J Safra Center for Bioinformatics at Tel-Aviv University (IGV, RW). IGV is a Faculty Fellow of the Edmond J Safra Center for Bioinformatics at Tel Aviv University and an Alon Fellow.

# Additional information

## Funding

| Funder | Grant reference number | Author |
| --- | --- | --- |
| Israel Science Foundation (ISF) | 1643/13 | Yael Oren, Roni Wilentzik |
| Israeli Centers for Research Excellence (I-CORE) | | Yael Oren, Roni Wilentzik |
| Tel Aviv University | Edmond J. Safra Center for Bioinformatics | Roni Wilentzik, Irit Gat-Viks |
| Broad Foundation | 1168/14 | Aharon Nachshon, Amit Frishberg |

The funders had no role in study design, data collection and interpretation, or the decision to submit the work for publication.

## Author contributions

YO, AN, IG-V, Conception and design, Analysis and interpretation of data, Drafting or revising the article; AF, RW, Analysis and interpretation of data, Drafting or revising the article

# Additional files

## Supplementary file

• Supplementary file 1. (**A**) Traits in the GEMOT modules of the BXD mouse strains. Shown is a module identifier (column 1) and details about a trait within it, including the trait title (column 2), PubMed record of the publication from which it is taken (column 3), the trait index in the GeneNetwork database (*Wang et al., 2003*; columns 4), and a short title of the trait as indicated in *Figure 3B* (column 5). (**B**) Drivers in the GEMOT modules of the BXD mouse strains. Shown is a module identifier (column 1), and details about a driver within it, including the gene symbol (column 2), its entrez identifier (column 3), genomic position (column 4), the type of association in the module (column 5), and its gene causality p value score (column 6). (**C**) Representative variants for causality testing in the tripartite modules of the BXD mouse strains. Shown is a tripartite module identifier and its GEMOT module identifier, if relevant (columns 1 and 2, respectively), the genomic position of the module's genomic interval (column 3), and the name and position of the representative variant (columns 4 and 5, respectively).

## Major datasets

The following previously published datasets were used:

| Author(s) | Year | Dataset title | Dataset ID and/or URL | Database, license, and accessibility information |
|---|---|---|---|---|
| Gerrits A, Li Y, Tesson BM, Bystrykh LV, Weersing E, Ausema A, Dontje B, Wang X, Breitling R, Jansen RC, de Haan G | 2009 | Expression quantitative trait loci are highly sensitive to cellular differentiation state | GSE18067; http://www.ncbi.nlm.nih.gov/geo/query/acc.cgi?acc=GSE18067 | Publicly available at NCBI Gene Expression Omnibus (http://www.ncbi.nlm.nih.gov/geo/). |
| Wang J, Williams RW, Manly KF | 2003 | WebQTL: web-based complex trait analysis | http://www.genenetwork.org/webqtl/main.py | BXD Trait Collection. Publicly available at Genenetwork. |

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
