## [Decision Letter]

Thank you for sending your work entitled “Linking traits based on their shared molecular mechanisms: A systems phenomics approach” for consideration at *eLife*. Your article has been favorably evaluated by Aviv Regev (Senior editor), a Reviewing editor, and 2 reviewers.

The Reviewing editor and the reviewers discussed their comments before we reached this decision, and the Reviewing editor has assembled the following comments to help you prepare a revised submission.

There is an overall appreciation for methods, such as the proposed GEMOT, that predict causative mutations leading to transcript variations that in turn drive phenotypes. However, to convince that the present method provides an advantage over existing approaches, three additions are required:

1) GEMOT performance should be compared to existing methods. This includes the qtlnet software (PMID 23288936, 21218138, and others) as well as similar methods (PMID 19540336, 21310061, 21242536, 25144184, 15711545, 25114278, 24013639, etc.).

2) Simulations should be used to evaluate the method against a 'truth' standard.

3) With regards to the *Klf7* results: a consistent model describing the effect of each allele (B6, DBA, overexpression, knockout) should be presented to provide a clear hypothesis of how *Klf7* effect is generated. (The interpretation of the *Klf7* effects is unclear in the manuscript. Firstly, what is the nature of variation between the two *Klf7* alleles? It has a cis-associated eQTL so, for example, which strain shows higher expression and can be linked to behavioral response to morphine? In the perturbation experiments, what is the strain background? Are the *Klf7* knockout and overexpression effects on “driver genes” (up or down-regulation) consistent with corresponding *Klf7* expression differences in the BxD population? That is, do *Klf7* knockout effects look like a low-expressing *Klf7* strain, and do *Klf7* overexpression effects look like a high-expressing *Klf7* strain? Figure 4 is also confusing as it appears to combine the knockout and overexpression experiments into summary scores. It would be useful to clearly see how these data support the inferred BxD model.)

4) The manuscript would be more convincing if the authors identify a group of traits that show low Pearson correlation among themselves, but share the same driver genes.

5) Considering the many methods already available (see above), the motivation for the present study should be better clarified.

6) The manuscript can be difficult to read at times, primarily due to a reliance on its own jargon. In some cases new nomenclature might be necessary, but many of the new terms seem to be very similar (if not identical) to widely-used, existing terms. In such cases, the paper should conform to standard nomenclature or define why such standards will not suffice. Notable examples are: “variant-gene associations” instead of eQTL; “linked relationships” instead of Pearson correlation and “link potential” instead of averaged correlation; “bipartite module” instead of pleiotropy. Furthermore, “gene drivers” and the “drivers layer” can probably be more simply referred to as “transcripts” and a “transcript layer” with appropriate context. “Causality score” appears to be a *P* value; why not refer to it as such? In some cases the new nomenclature can be potentially misleading, such as in the *Klf7* discussion where a *P* value is referred to as a “perturbation effect”. Effect size and significance are different concepts with different biological interpretations.

7) High-dimensional approaches like GEMOT are prone to generating false positive associations, especially in such small experimental populations. Are *P* values systematically corrected for multiple tests? It is also important to estimate false discovery rates for a given significance threshold, which are not addressed in the manuscript. Additionally, the sample size in this study is very small, and BxD lines often exhibit non-local linkage disequilibrium due to limited recombination across few individuals. This can lead to many false-positive associations. How has this potential issue been addressed in the current study?

---

## [Author Response]

*There is an overall appreciation for methods, such as the proposed GEMOT, that predict causative mutations leading to transcript variations that in turn drive phenotypes. However, to convince that the present method provides an advantage over existing approaches, three additions are required*:

*1) GEMOT performance should be compared to existing methods. This includes the qtlnet software (PMID 23288936, 21218138, and others) as well as similar methods (PMID 19540336, 21310061, 21242536, 25144184, 15711545, 25114278, 24013639, etc.)*.

We thank the reviewers for their suggestion, which we believe improves our conclusions. The specified publications are focused on three different aspects: (1) Testing causative relationships (Neto 2013, Lee 2009); (2) Network reconstruction (Neto 2010, Hageman 2011, Wang 2014); and (3) grouping traits (Chesler 2005). As suggested by the reviewers, in the revised manuscript we added simulations that relate to each of these three aspects, with a focus on comparison to the suggested methods.

(1) Testing causative relationships. We agree with the reviewers that our original manuscript did not provide assessment of our causality testing methodology. Notably, our goal was not to develop a new method for a causality *P*-value score. Rather, we aimed to use an existing methodology with minor modifications that were required in our particular case. In accordance, we used a methodology that is very similar to Neto et al. 2013, with two small modifications: first, we use empirical rather than theoretical evaluation of the likelihood ratio distribution; second, Neto et al. was interested in a 'classical' definition of causality, where a transcript translates between a variant and a trait. Here, we were interested in an additional objective function: identifying causal transcripts in the presence of other transcripts. This situation is common in our case of GEMOT modules harboring multiple causal transcripts translating between the same variant and a group of traits. We have therefore slightly modified the method suggested by Neto et al. to capture this 'broad' definition of causality. We apologize for omitting this important information from the original submission. We now added a detailed description of the methodology and alterations from Neto et al. to the subsection headed “GEMOT stage III: Refine module (validate drivers). We also added a new simulation study (new Figure 1—figure supplement 2, subsection “Causality testing—performance evaluation”, legend for Figure 1—figure supplement 2), which shows that in comparison to the compared methods (Neto et al. and Lee et al.), GEMOT attained improved accuracy in our relevant case of a broad-sense definition of causality. We now clarify this in the Methods section.

(2) Network reconstruction. In the third stage of the GEMOT algorithm, GEMOT aims to identify the groups of traits that share causal drivers within the given tripartite modules. Such identification can also be performed by constructing a detailed network within a module (using existing methods) and using this solution to identify the desired group of traits. Notably, our goal was not to better construct any given network. Rather, we aimed to develop scalable methodology for the identification of the shared drivers of the module's traits, which can scale to the large tripartite modules we found in real mouse data (with a typical size of over 100 candidate transcripts and 2-40 traits in the identified tripartite modules). According to our observations, the accuracy of the existing methods dropped drastically with an increased number of components, and had an unrealistic running time for our BXD modules. This led us to develop an alternative methodology in GEMOT phase III for identifying the players behind the group of traits without a full reconstruction of the network. However, the GEMOT pipeline is general and existing network reconstruction methods may be used in phase III of the pipeline, alongside or instead of our suggested phase-III approach. We now clarify these points in the Introduction, Results and Methods sections.

In the revised manuscript, we added a new simulation study that takes as input co-mapped subnetworks that mimic tripartite modules so as to identify the desired group of traits that share the same transcript players behind them. Notably, GEMOT does not construct a causal network but only predicts a subset of traits sharing the same mechanism. To avoid biases, we therefore based our systematic evaluation on the ability to identify the correct group of traits sharing a mechanism, thus ignoring irrelevant additional information produced by the compared network reconstruction methods. We now provide a detailed description of the simulated sub-networks and comparisons in the Results and Methods (subsections “Systematic evaluation of the GEMOT algorithm using synthetic data analysis” and “GEMOT Performance analysis”) and new five figures (new Figure 5 and new Figure 5—figure supplement 1, Figure 5—figure supplement 2, Figure 5—figure supplement 3 and Figure 5—figure supplement 4). These simulation results support our usage of GEMOT in the range of biologically-relevant network parameters.

(3) Grouping traits. We have added a relevant simulation of a large biological network that harbors small groups of traits that share the same causal transcripts within it. We now discuss the simulation of this large network in the Methods (“GEMOT Performance analysis”) and Results (“Systematic evaluation of the GEMOT algorithm using synthetic data analysis”) sections. GEMOT's performance is compared to two alternative methods: (I) the ICE algorithm (52), which was used by Chesler et al. 2005 (as recommended by the reviewers); and (II) the INVAMOD algorithm. We find that GEMOT outperforms the alternative methods in identifying groups by causality across varying numbers of traits in a group (new Figure 5—figure supplement 5).

Importantly, an equivalent systematic comparison between GEMOT and the alternative network reconstruction algorithms on the entire network is not feasible due to the long running time of the network reconstruction algorithm. Therefore, the entire 3-phase GEMOT pipeline, which can be applied on thousands of components, cannot be directly compared to the network reconstruction methods. We now highlight this point and exemplify the running time of the compared methods for the case of small sub-networks (new Figure 5—figure supplement 2).

Notably, three additional suggested publications (Westra 2013, Rutledge 2014 and Rosa 2009) have not been directly compared since they do not provide or develop any code that we could systematically apply to the synthetic data; these publications, however, discuss methodologies that are similar to the methods by Chesler 2005, Neto 2013 and Lee 2009, which were compared in our revised manuscript, as detailed above.

*2) Simulations should be used to evaluate the method against a 'truth' standard*.

We addressed this issue above (in our answer to comment 1). As noted, we now added simulations to evaluate GEMOT's accuracy. In particular, using a gold standard, we present the superiority of GEMOT for the goal of identifying groups of traits with causative players behind them. This is shown for synthetic networks of increasing complexity:

(I) Small network fragments including triplets of components (new Figure 1—figure supplement 2, subsection “Causality testing—performance evaluation” and Figure 1—figure supplement 2, legend).

(II) Co-mapped sub-networks (new Figure 5 and Figure 5—figure supplement 1, Figure 5—figure supplement 2, Figure 5—figure supplement 3 and Figure 5—figure supplement 4, and subsections “GEMOT Performance analysis” and “Systematic evaluation of the GEMOT algorithm using synthetic data analysis”).

(III) Full large networks (in the same subsections).

*3) With regards to the* Klf7 *results: a consistent model describing the effect of each allele (B6, DBA, overexpression, knockout) should be presented to provide a clear hypothesis of how* Klf7 *effect is generated. (The interpretation of the* Klf7 *effects is unclear in the manuscript. Firstly, what is the nature of variation between the two* Klf7 *alleles? It has a cis-associated eQTL so, for example, which strain shows higher expression and can be linked to behavioral response to morphine? In the perturbation experiments, what is the strain background? Are the* Klf7 *knockout and overexpression effects on “driver genes” (up or down-regulation) consistent with corresponding* Klf7 *expression differences in the BxD population? That is, do* Klf7 *knockout effects look like a low-expressing* Klf7 *strain, and do* Klf7 *overexpression effects look like a high-expressing* Klf7 *strain?*
Figure 4
*is also confusing as it appears to combine the knockout and overexpression experiments into summary scores. It would be useful to clearly see how these data support the inferred BxD model*.*)*

Following the reviewers' suggestion, we first extended the analysis of module number 2 using the BxD data, and then demonstrated the agreement between this data and the perturbation experiment. In the revised manuscript we present a full consistency between the experimental perturbation results and the BxD-based model. Our extended analysis shows that the drivers in module number 2 can be partitioned into two types: group I is positively correlated with the *Klf7* transcript and negatively correlated with the morphine-related traits; group II is negatively correlated with *Klf7* and positively correlated with these traits. Consistently, strain DBA/2J shows higher levels of group I and *Klf7* transcripts, whereas strain C57BL/6J shows higher levels of group II transcripts and of the morphine-related trait measurements. These results suggest that *Klf7* acts as an activator of group I drivers and as a repressor of group II drivers. In accordance with this prediction, *Klf7* overexpression in the C57BL/6J strain background up-regulated group I and down-regulated group II. Furthermore, *Klf7* knockdown in the C57BL/6J strain background down-regulated group I, as expected. (*Klf7* knockdown have no influence on group II, likely due to the low *Klf7* level in the wild-type C57BL/6J, which already leads to an ineffective repression). We thank the reviewers for this suggestion, which we believe strengthened our results.

We have substantially revised the manuscript to include these new results that now support a causal role for *Klf7* as an activator of drivers in group I and repressor of drivers in group II (in “Discovery of a model for a collection of tightly related traits”, in Results), and present new Figure 3–group I and new Figure 3—figure supplement 1, Figure 3—figure supplement 2 and Figure 3—figure supplement 3 (which replaces former Figure 4 and Figure 4—figure supplement 1) by providing all the relevant data about the partition of drivers into two groups, the correlation between objects and the difference in trait and expression levels between alleles, along with the corrected final model. In particular, we do not use a summary score for overexpression and knockdown, as suggested by the reviewers. Instead, Figure 3 now present the results of over-expression and knockdown in separate plots (replacing the unified presentation in former Figure 4).

*4) The manuscript would be more convincing if the authors identify a group of traits that show low Pearson correlation among themselves, but share the same driver genes*.

We agree with the reviewers, and apologize for the missing details in our description of the modules in the original submission. The original submission already exemplified two modules with high trait-trait correlations (modules nos. 2 and 3) and one module with poor trait-trait correlations (module number 4, in formerly Figure 5, currently Figure 4). The revised manuscript clarifies this point in the Results and Discussion sections. We also added a new Figure 2—figure supplement 4 presenting the trait-trait correlations in each of the modules.

*5) Considering the many methods already available (see above), the motivation for the present study should be better clarified*.

The comment pointed out to a lack of clarity in comparison to existing methods, which we now address in full. First, we clarify the difference from existing methods. Whereas many trait-clustering methods are available, these do not rely on causative molecular players behind the grouped traits. Furthermore, whereas many causal network construction methods exist, such methods construct the entire networks and therefore cannot scale to large networks that include thousands of genes and traits (due to a lack of power and a non-realistic running time). We have revised the text in the Introduction, Results , and Discussion sections to clarify these points. Second, in the revised manuscript, we use simulations to demonstrate the advantages of GEMOT over various existing methods (network construction: Neto 2010, Hageman 2011, Wang 2014; grouping methods: [52], [16]) for the goal of identifying groups of traits that share the same causal transcripts, comparing both running time and accuracy. In addition, we clarify the differences from existing causality testing approaches, including a comparative simulation study. We address the simulation studies in detail in response to comment 1, above. Third, we emphasize that the GEMOT pipeline is general and different existing methods may be used as subroutines within the GEMOT pipeline. For example, network construction methods may be applied to construct the sub-networks within the tripartite modules; existing causality-testing methods may be used to identify the driver genes; existing biclustering approaches may be utilized to identify the bipartite modules. We now clarify this point in the revised manuscript.

*6) The manuscript can be difficult to read at times, primarily due to a reliance on its own jargon. In some cases new nomenclature might be necessary, but many of the new terms seem to be very similar (if not identical) to widely-used, existing terms. In such cases, the paper should conform to standard nomenclature or define why such standards will not suffice. Notable examples are: “variant-gene associations” instead of eQTL; “linked relationships” instead of Pearson correlation and “link potential” instead of averaged correlation; “bipartite module” instead of pleiotropy. Furthermore, “gene drivers” and the “drivers layer” can probably be more simply referred to as “transcripts” and a “transcript layer” with appropriate context. “Causality score” appears to be a* P *value; why not refer to it as such? In some cases the new nomenclature can be potentially misleading, such as in the* Klf7 *discussion where a* P *value is referred to as a “perturbation effect”. Effect size and significance are different concepts with different biological interpretations*.

We thank the reviewers for this suggestion, and made the following changes accordingly:

1) Transcripts: The term 'gene' was replaced with ‘transcript’, and accordingly, the terms ‘driver layer’ and 'molecular layer' were replaced with ‘transcripts layer’. We modified the text throughout the manuscript accordingly.

2) Driver-related terms: We simplified the terminology by replacing the term 'candidate driver' with 'candidate transcript', and 'driver' with 'driver transcript'. Notably, we could not completely replace the term 'driver' and use the term 'transcript' instead, since we have two kinds of transcripts: the candidate transcripts in stage number II and the transcripts that were identified as drivers in stage number III.

3) Causative scores: The term 'causative score' was replaced with 'causality *P* value', and the term 'gene causality score' was replaced by 'transcript causality *P* value', as suggested.

4) The *Klf7* discussion: In the results referring to module number 2 we now conformed to the standard terminology, where 'effect size' refers to difference between strains, and 'significance of effect' refers to the *P* value for these differences.

5) Linked relationships and link potential: In both cases, the term could not be replaced with an existing term. In the case of 'Linked relationships', we believe that replacing it with the term 'correlation' is misleading since 'correlation' refers to relations between two objects, whereas 'linked relationships' refers to the relations between three objects. In particular, 'linked relationships' refers to the combination of *two* existing terms: one is a simple 'correlation' between a transcript and a trait, and the second is an 'association' of a variant and a transcript. Therefore, there is no existing term referring to this combination. We now have the text slightly refined accordingly. Notably, since the term 'link potential' refers to an aggregation of 'linked relationships', we therefore could not use an existing term in this case.

6). Bipartite modules: In the GEMOT pipeline, bipartite modules do not represent any kind of pleiotropy, but only pleiotropy that is likely mediated through transcripts. The term 'Bipartite module' is therefore not equivalent to 'pleiotropy' and cannot be replaced. We now clarify this in the revised manuscript. Notably, bipartite graphs and sub-graphs conform to the standard terminology in the biclustering literature (*e.g.*, Tanay et al., Bioinformatics 2002).

7) Variants: In some cases along the GEMOT pipeline, using the term 'eQTL' instead of 'variant' may be misleading. In particular, for the output GEMOT modules and association scores, the variant can be also terms an eQTL. However, in the remaining manuscript, the variant is not an eQTL. For example, during phase 1 of GEMOT, the variant only has linked relationships with a trait, therefore it cannot be declared as an eQTL; in bipartite and tripartite modules and also throughout the simulation studies (Figures 1 and 5), reactive relationships may appear, and therefore the variant is not an eQTL. We believe that a consistent usage of the term 'variant' in all cases is more clear than using two different terms for the same object in different contexts.

Overall, in the revised manuscript, we changed 8 terms so as to conform to the standard terminology according to the reviewers’ suggestion.

*7) High-dimensional approaches like GEMOT are prone to generating false positive associations, especially in such small experimental populations. Are* P *value systematically corrected for multiple tests? It is also important to estimate false discovery rates for a given significance threshold, which are not addressed in the manuscript. Additionally, the sample size in this study is very small, and BxD lines often exhibit non-local linkage disequilibrium due to limited recombination across few individuals*. *This can lead to many false-positive associations. How has this potential issue been addressed in the current study?*

The reviewer raises two concerns regarding false positives. The first refers to the evaluation of FDR and the multiple testing problem, and the second relates to the linkage disequilibrium issue.

1) Multiple testing correction and FDR: In the revised manuscript, we now systematically correct for multiple testing and estimate FDR. First, in the case of real data, FDR is estimated based on a permutation test. We assessed the FDR for various predictions along the GEMOT pipeline, including the thresholds for identifying candidate transcripts and driver transcripts. We have substantially revised the Methods section accordingly. Second, we provide direct evaluation of FDR for the grouping of traits in synthetic networks, when the actual solution is known (new Figure 5—figure supplement 5 and Figure 1—figure supplement 2). Finally, We now systematically added multiple testing corrections for all reported *P* values throughout the manuscript. Notably, GEMOT provides the same results when applied with or without Boneferroni correction for the gene-variant association and link potential scores, since the construction of the bipartite modules is not sensitive to scaling of these values. We clarify this in the revised text.

2) Non-local linkage disequilibrium: We certainly agree that in some cases it would not be possible to discern between several potential genomic intervals due to non-local linkage disequilibrium. To address this issue, we tested the linkage disequilibrium (LD) between the genomic interval of each of our identified modules and the entire genome. We added a new Figure 2—figure supplement 1 presenting these results, allowing the observation of additional potential intervals for a module. We briefly discuss it in the manuscript (please see the Discussion section and the legend for Figure 2—figure supplement 1). For the particular modules presented in this study, all non-local LDs did not exceeded |r|=0.75.